

# Impacts of topographic shading on direct solar radiation for valley glaciers in complex topography

Matthew H. Olson, Summer B. Rupper

Department of Geography, University of Utah, Salt Lake City, 84112, USA

*Correspondence to*: Matthew H. Olson (matthew.olson@geog.utah.edu)

**Abstract.** Topographic shading, including both shaded relief and cast shadowing, plays a fundamental role in determining direct solar radiation on glacier ice. However, this parameter has been oversimplified or incorrectly incorporated in surface energy balance models in some past studies. Here we develop a topographic solar radiation model to examine the variability in irradiance throughout the glacier melt season due to topographic shading and combined slope and aspect. We apply the

model to multiple glaciers in High Mountain Asia (HMA), and test the sensitivity of shading to valley-aspect and latitude. Our results show that topographic shading significantly alters the potential direct clear-sky solar radiation received at the surface for valley glaciers in HMA, particularly for north- and south-facing glaciers. Additionally, we find that shading can be extremely impactful in the ablation zone. Cast shadowing is the dominant mechanism in determining total shading for valley glaciers in parts of HMA, especially at lower elevations. Although shading has some predictable characteristics, it is overall

extremely variable between glacial valleys. Our results suggest that topographic shading is not only an important factor contributing to surface energy balance, but could also influence glacier response and mass balance estimates throughout HMA.

## 1   Introduction

Valley glaciers are an important resource for many local communities, where summer melt is vital for irrigation and drinking sources (Immerzeel et al., 2010). Additionally, after thermal expansion, mountain glaciers are expected to be the next largest

contributor to sea-level rise over the next century (Vaughan et al., 2014). Improvement to our understanding of these glacial systems is essential in properly quantifying melt and associated impacts, particularly in remote regions where in situ data is sparse.

During the summer months, net shortwave radiation on a glacier is one of the main components of surface energy balance, often accounting for 75% or more of available energy at the surface (Gruell & Smeets, 2001; Oerlemans and Klok, 2002).

Thus, changes in the amount of solar radiation at the surface will alter the overall global radiation, and consequently be a significant influence on surface energy balance. The intensity of solar radiation received at the surface of a glacier is primarily a function of latitude and time of year, with parameters such as topographic shading, slope, and aspect controlling the distribution of radiation on a local scale. Despite the potential importance of topographic shading, this process is often over-simplified in glacier energy balance calculations due in part to computational expense and added complexity.



To facilitate further discussion and analysis, we separate topographic shading into two components: shaded relief and cast shadow. Shaded relief occurs due to self-shadowing at a given location and relies solely on slope and aspect. If the angle between the solar position and surface normal at a given location is greater than 90°, it is treated as "shadow," if less, it is considered "in sun," as seen in Fig. 1. This method is also often used as an image enhancing technique for display purposes, also commonly described as a hillshade (e.g. ESRI Hillshade Tool).

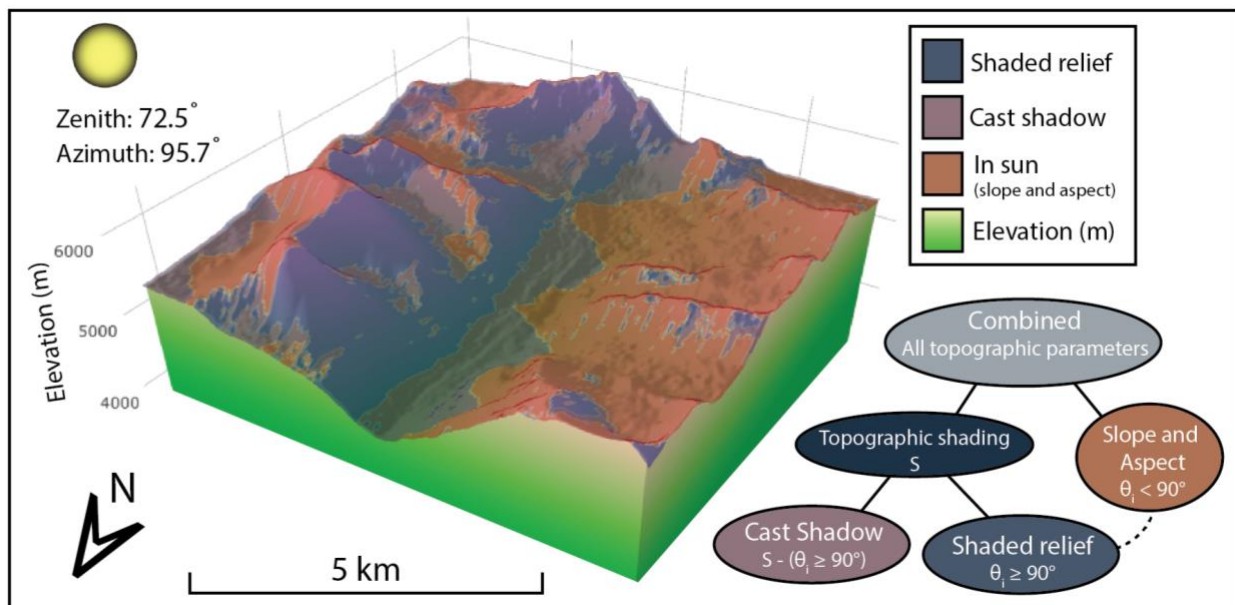

**Figure 1. Comparing two different shading methods in the Satluj sub-basin within the Indus watershed. Shading is calculated on April 1, 2013 at 7:33am. Notice the importance of incorporating both methods in order to determine shading in this glacial valley. Both methods, but specifically cast shadows, are most prevalent early in the morning and late in the evenings when the zenith angle is large. Symbols in the flow chart relate to equations in section 2, $\theta_i$ denotes the incidence angle.**

Cast shadows are the result of shading due to adjacent topography. Nearby features, such as surrounding valley walls, may block the neighbouring area from direct solar radiation, as depicted in Fig. 1, particularly in the early and late hours of the day. This effect can be more pronounced near steep terrain and narrow valleys. Because local terrain is unique and solar angles vary constantly, cast shadows must be considered for each sun position over an hourly, daily, and monthly time interval (Aguilar et al., 2010).

With the global availability of 30-meter, and finer, resolution digital elevation models (DEMs), slope and aspect can be easily incorporated in solar radiation calculations. However, many radiation models used in glacier studies only incorporate shaded relief without including cast shadowing (Chen et al., 2013; Han et al., 2016; Hopkins et al., 2010; Kumar et al., 1997; Plummar et al., 2003; Zhang et al., 2015; and others). As such, multiple studies have interpreted the impact of topographic shading as being negligible to the overall radiation budget. While this leads to an incomplete inclusion of true topographic shading, there are many important exceptions that model this parameter correctly (e.g., Aguilar et al., 2010; Arnold et al.,





2006; Dozier et al., 1990; Hock, 1999; Klok and Oerlemans, 2002; Williams et al., 1972). For example, Klok and Oerlemans (2002) show that topographic shading can reduce shortwave radiation by more than 10 % at lower elevations, and the combined effect of topography can result in a 37 % overestimation of incoming solar radiation. Munro et al. (1982) also determined that shading is generally greatest at lower elevations in a glacier basin, but suggested that the overall impact is negligible on the

5    energy budget. Alternatively, Arnold et al. (2006) observed a significant effect and general increase in shading with elevation, as well as a large impact near valley walls for an Arctic glacier. Additionally, he suggested that the influence of topography on surface energy balance would be enhanced at higher latitudes. Other studies have indicated that direct solar radiation can be obstructed on certain steep mountain slopes for most of the year (Aguilar et al., 2010). These results suggest that large errors can be introduced when modelling net solar radiation if topography is not correctly incorporated. However, further insight is

10   needed to better identify the magnitude and spatial patterns of topographic shading on glaciers in complex terrain.

This study focuses on improving the current understanding of both shaded relief and cast shadowing on glaciers in the complex and varied topography of High Mountain Asia (HMA) (Fig. 2). We compare the impact of these two shading methods to the collective influence of slope and aspect, as they are generally assumed to be the dominant topographic factors affecting solar radiation in mountainous terrain (Dozier et al.,1980).

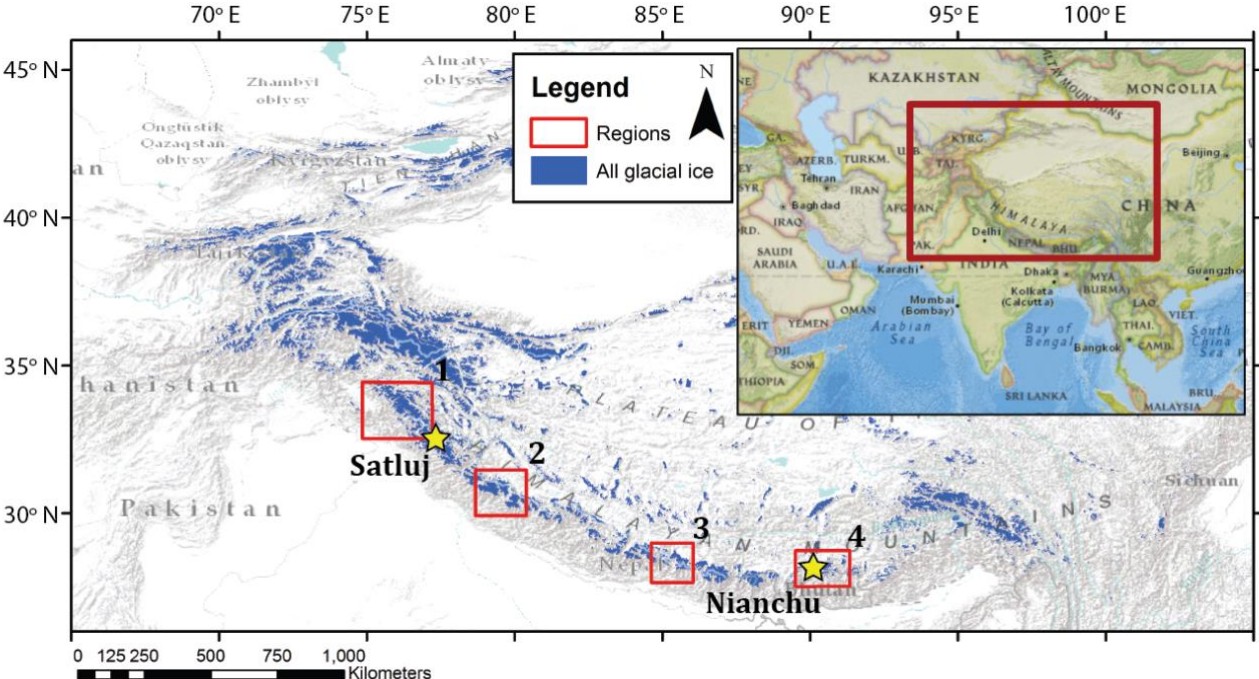

**Figure 2. High Mountain Asia, showing glacier ice in blue. Two selected glaciers of interest are shown as yellow stars. Four red boxes indicate the areas selected for regional analysis. Regions are centered on the Jammu Kashmir (1), Himarchal Pradesh (2), Everest (3), and Bhutan (4) regions. Glacier shapefiles from Arendt et al., 2015, Basemap acquired from ESRI, March 2017.**



## 2  Data and methods

In this study, we will use a solar radiation model in conjunction with a topographic model to simulate the distribution of direct solar radiation across glacier surfaces. The models will be used to parse out the change in irradiance due to specific topographic parameters such as slope and aspect, shaded relief, and cast shadows. The key inputs to the models will be glacier location and size, and digital elevation models (DEMs).

First, we apply these two models to two individual glaciers in HMA. We then test the sensitivity of each glacier to different valley-aspects, and latitudes using idealized scenarios. Finally, we evaluate variations in topographic shading across four glacierized basins that represent distinct zones of differing latitude, topography, and climatology within HMA.

### 2.1  Modelling solar radiation

Accurately determining the position of the sun is essential in order to properly calculate the amount of incoming solar radiation. This is also necessary when considering topographic effects. Assuming a flat plane, the solar position is described by a combination of the zenith (Z) and azimuth angles which are calculated using standard methods (Iqbal, 2012).

We model potential clear-sky direct solar radiation (Ia), as it passes through the atmosphere, at 15-minute time-intervals throughout the melt season (April 1—September 31):

$$I_a = I_0 \left( \frac{R_m}{R} \right)^2 \psi_a^{\frac{P}{P_0 \cos Z}} \tag{1}$$

where $I_0$ is the solar constant (~1368 Wm$^{-2}$), R is the sun-Earth distance (subscript m refers to mean), $\psi_a$ is atmosphere clear-sky transmissivity (a constant of 0.75 is used (Hock, 1999)), P is atmospheric pressure calculated using a simple lapse rate for dry air, $P_0$ is mean atmospheric pressure at sea level, and Z is local zenith angle (which accounts for the size of air mass that radiation must travel through before arriving at the surface). Equation (1) has been modified from Hock (1999) to exclude the parameter responsible for attenuation at the surface. This term will be added later in the topographic model. The secant simplification for estimating air mass in Eq. (1) over-estimates atmospheric attenuation with zenith angles larger than 70° (Williams et al., 1972). Because shading is most significant during high zenith angles, an improved approximation of air mass would likely provide a larger estimate of daily mean changes in potential clear-sky direct irradiance due to topographic shading. Consequently, our results will be conservative estimates.

### 2.2  Topographic modelling

In alpine terrain, the combination of slope and aspect are generally presumed to be the most influential topographic factor regulating absorbed solar radiation at the surface (Dozier et al. 1990). However, some areas surrounded by steep terrain can also be highly influenced by topographic shading (Arnold et al., 2006). Due to the high spatial and temporal variability in the incident angle and topographic shading throughout the day, comparing the impact of these topographic parameters side-by-





side to assess relative importance can be challenging. Here we present Eq. (3-6) as a means to address this issue. We use these equations to calculate the change in irradiance averaged over the course of a day for a given topographic parameter.

We incorporate two additional terms, incident angle and topographic shading, in conjunction with Eq. (1) to determine the distribution of solar radiation at the surface due to topographic effects. Potential clear-sky solar radiation arriving on an inclined surface is:

$$I_c = I_a cos\theta_i S \tag{2}$$

where Ia is potential atmospheric clear-sky direct solar radiation from Eq. (1), $\theta_i$ is the incident angle, and S is topographic shading, a binary value indicating whether a given cell is in "shade" (0) or "sun" (1). Topographic shading is calculated with a modified ray-tracing algorithm that uses a solar illumination plane perpendicular to the sun's zenith angle in order to determine if a cell is blocked by surrounding cells at a certain zenith and azimuth angle (Corripio, 2003). This method incorporates both self-shading from relief and cast shadows. The incident angle is the zenith angle (Z) modified for a surface with a specific slope and aspect (Iqbal, 2012).

We use variations of Eq. (2) in order to determine the daily mean change in solar radiation from specific topographic parameters shown in Fig. 1. Equations (3-6) show the mean change in solar irradiance due to slope and aspect (incidence angle), shaded relief, cast shadows, and the combined effect of these topographic parameters:

$$\overline{\Delta I_{SA}} = \frac{1}{t}\int_1^t I_a \cdot S\,[\cos\theta_i - cosZ]\,dt \tag{3}$$

$$\overline{\Delta I_{SR}} = \begin{cases} \dfrac{1}{t}\displaystyle\int_1^t -I_a \cdot cosZ, & \theta_i \geq 90° \\ \\ \qquad\qquad 0, & \theta_i < 90° \end{cases} \tag{4}$$

$$\overline{\Delta I_{CS}} = \frac{1}{t}\int_1^t I_a \cdot \cos\theta_i\,[S-1]\,dt \tag{5}$$

$$\overline{\Delta I_{Com}} = \frac{1}{t}\int_1^t I_a\,[S \cdot \cos\theta_i - cosZ]\,dt \tag{6}$$

Equation (3) shows the daily mean change in solar radiation due to the slope and aspect relative to a flat plane. By incorporating S in this equation, values considered to be in shade are excluded. The mean change in irradiance due to shaded relief is also relative to a flat plane (Eq. (4)), as shaded relief also relies on the presence of slope and aspect values. Shaded relief only occurs when the incident angle is greater than or equal to 90°. Equation (5) shows the mean change in irradiance





due to cast shadowing relative to an inclined plane void of cast shadows. This accounts for any change in irradiance due to the slope and aspect, as well as shaded relief. By default, shaded relief is incorporated in both the calculation of the incident angle ($\theta_i$) and the shading algorithm used to determine topographic shading (S). This allows us to parse out the individual impact from each of these topographic parameters, for comparison. Finally, Eq. (6) is the combined effect from these three parameters

relative to a flat plane. For each of the equations described above, the mean change in irradiance is equivalent to the integrated difference over the course of a day.

Our results present a daily mean change in irradiance due to each topographic parameter, averaged across the entire melt season. For simplicity, we will refer to the daily mean change in irradiance averaged over the melt season as the mean change in irradiance.

## 2.3 Data

This study utilizes the 30-meter resolution ASTER global digital elevation model (GDEM) to simulate topographic terrain in the models. Although a 30-meter resolution DEM does not fully capture the actual topographic complexity in a glacial valley, it serves as an adequate representation in order to measure the overall effect of surrounding topography on solar radiation within our theoretical framework. DEM resolution is further considered in the discussion section of this paper.

Glacier boundaries are determined with the latest shapefiles available from the Randolph Glacier Inventory 5.0 and ICIMOD (Arendt et al., 2015; Bajracharya and Shrestha, 2011). Both inventories delineate glacier ice using a variety of techniques to determine glacier boundaries. A buffer of 5 km is generated around the glacier shapefiles in order to include all surrounding topography. Because valley glaciers are constrained by the immediate surrounding topography, only nearby features are able to affect incoming solar radiation. However, at higher glacier elevations, the visible horizon can become much larger, in which

case the extent must incorporate topographic features within visibility. For this study, a buffer of 5 km proved to be sufficient. For example, topographic shading was altered by less than 0.01% when changing DEM extent from 5 km to 3 km beyond the Satluj glacier boundary.

## 2.4 Idealized scenarios and regional analysis

We implement two idealized scenarios in order to test the sensitivity of the shaded relief and cast shadows to changes in overall

glacier aspect and latitude. The components of topographic shading are calculated for two different glaciers as they are rotated in each of the main cardinal direction (North, East, South, and West). This allows us to observe changes in direct irradiance due to different valley aspects for the two glaciers. We then calculate the relative change in irradiance due to topographic shading across varying degrees of latitude (20-50°) for the same two glaciers. Glacier size, slope, and surrounding topography are constant in these idealized scenarios, while aspect and latitude are systematically varied.

We also apply our cast shadow model to four glaciated regions across the greater Himalaya, shown in Fig. 2. These regions span multiple latitudes and degrees of topographic relief, and include a larger sampling of glacier geometries throughout HMA. Only glaciers larger than 3 km2 were used in the analysis in order to exclude small cirque glaciers, and focus on valley glaciers



with a developed tongue. These glaciers are then grouped based on general valley aspect of the ablation zone (ice below mid elevation), as we are most interested in the cast shadowing effect from the valley walls along lower elevations where melt is dominant. The valley aspect for each glacier is determined based on the mode of pixels of a reclassified aspect value. They are then grouped into two categories: North/South and East/West oriented glaciers. Glaciers with less than 10% majority aspect

are manually corrected and labelled according to the best perceived aspect. This introduces some complication as glacier aspect and morphology can be extremely irregular throughout the lower elevation for these HMA glaciers. For this reason, we include a large sample of valley glaciers for each region, ranging from 80 to 117 glaciers. A single mean value of change in solar radiation due to topographic shading is calculated across the defined ablation zone of each glacier. A kernel density estimation is fit to the mean values in each region for both North/South and East/West glaciers in order to see how the distribution of each

group varies from one another.

## 3   Results and discussion

### 3.1   Topographic impacts on individual glaciers

We selected a glacier in the Panjnad basin of the Indus watershed in the Himarchal Praedesh region of the western Himalaya, and one located in the eastern Himalaya on the China-Bhutan border, for our detailed analysis and idealized scenarios. We

refer to the glacier in the western Himalaya as the Satluj Glacier, due to the sub-basin in which it resides. Similarly, we refer to the glacier in the eastern Himalaya as Nianchu Glacier, after its sub-basin name. We chose these two valley glaciers because they both have clear, north-facing aspects, and well-developed glacier tongues. However, the Satluj glacier is in an area of steep topography and high relief, while Nianchu Glacier spans a slightly higher range of elevation and is surrounded by less topographic relief.

### 3.1.1   Satluj Glacier

We calculate daily mean change in irradiance due to each topographic parameter averaged throughout the summer melt season (April 1–September 31). Figure 3 shows a smoothing spline of the mean change in irradiance due to each topographic parameter and their totals across elevation for the Satluj Glacier. The spatial variability across the glacier surface is also included for visual aid (Fig. 3a–d). The mean change in irradiance for all combined topographic parameters is greatest at the

lowest elevations (< 4800 m), where cast shadowing is the largest, and at higher elevations (5600 m), where the impact of slope and aspect is largest. Cast shadowing only plays a significant role where the surrounding terrain is steep, close in proximity, and large enough to cast a significant shadow over the valley. For this particular glacier, this occurs at the lower elevations of the ablation zone. By comparison, slope and aspect tend to be important across all elevations but become slightly more significant as elevation increases. The effects of shaded relief on irradiance also increase with increasing elevation.





**Figure 3. Satluj Glacier.** Mean change in irradiance throughout the summer melt season due to shaded relief (a), cast shadowing (b), slope and aspect (c), and combined topographic parameters (d). A smoothing spline was fit to the irradiance change values along the elevation profile of the glacier. The effect of cast shadowing shows to be more significant than that of shaded relief and slope and aspect at lower elevations where ablation dominates during the melt season.





Above 5050 m, the glacier cirque spreads out to incorporate various aspects. These aspects receive more direct solar radiation than their north-facing counterparts (Fig. 3c–d). Additionally, the mean change in irradiance from cast shadowing significantly decreases in both the east- and west-facing aspects, but not on a small south-facing tributary (Fig. 3b–right side). Change in irradiance on glacier ice less than 5050 m is clearly dominated by cast shadowing. This result is critical, as an overestimation of net shortwave radiation in the ablation zone (e.g., where the energy budget is frequently already positive) will have a larger effect on glacier mass balance than an overestimation in the accumulation zone (e.g., where the energy budget is usually negative enough that a small increase in incoming energy likely won't result in significant melt). Furthermore, Fig. 3 shows that the influence of topographic shading is also more significant than the mean change in irradiance from slope and aspect throughout the ablation zone. However, this is not true when comparing the effects of slope and aspect with shaded relief alone. These results emphasize the importance of correctly incorporating both methods of topographic shading when modelling solar radiation or surface energy balance, particularly for north-facing valley glaciers in areas of high relief.

### 3.1.2 Nianchu Glacier

Figure 4 shows an overall increasing mean change in irradiance from combined topographic parameters with increasing elevation for the Nianchu Glacier. Both shaded relief and slope and aspect follow a similar trend of increasing mean change in irradiance with elevation, similar to the Satluj Glacier. However, in contrast to Satluj, cast shadowing on Nianchu tends to have only a slightly larger effect than slope and aspect at elevations below 5500 m.

The lower impact from cast shadows on the lower elevations for the Nianchu is due to lower relief from the surrounding valley walls alongside the glacier toe, as well as the slight north-eastern direction of the valley. Sunrise occurs further northeast during the summer months than during the rest of the year, and if the valley faces this direction and is surrounded by lower relief, the effect from cast shadowing can be minimal. Once above 5500 m, cast shadowing increases dramatically (Fig. 4), then gradually decreases again at higher elevations. This shadowing can be seen in Fig. 4b, at the point in which the glacier divides into two tributaries and becomes more north facing. Although the Nianchu Glacier shows how variable topographic shading can be, shading is still a significant influence on the overall mean change in irradiance at the surface, even surpassing the impact of slope and aspect at certain elevations. However, because topography is highly variable and differs significantly from one basin to the next, the effects of topographic shading are equally heterogeneous.

### 3.2 Idealized scenarios

We perform idealized scenarios in order to quantify how topographic shading changes over different valley aspects and latitudes. In particular, we rotate the Satluj and Nianchu glaciers in each of the main cardinal directions, and move each glacier across varying degrees of latitude.





**Figure 4. Nianchu Glacier. Mean change in irradiance throughout the summer melt season due to shaded relief (a), cast shadowing (b), slope and aspect (c), and combined topographic parameters (d). A smoothing spline was fit to the irradiance change values along the elevation profile of the glacier. The effect of cast shadowing is largest at mid elevations and occurs once the Nianchu becomes more directly north-facing and is constricted between steep valley walls.**





Only slight changes in the shaded relief occur when the Satluj glacier is rotated in each cardinal direction. On the other hand, cast shadowing shows a significant mean decrease in irradiance at lower elevations for both north- and south-valley aspects, compared to east and west (Fig. 5). This is expected, as shading is largest when the sun is low in the sky and nearby topography is oriented such that it blocks direct solar rays. This occurs during the early and late hours of the day when large terrain features are situated to the east and west. East- and west-valley aspects lose the orientation of steep, adjacent topography able to cast significant shadows, thus the impact is greatly reduced. However, we see an increase in cast shadowing in the upper elevations for east and west aspects, as the upper tributaries in Fig. 3 are now predominantly facing north and south.

The Nianchu Glacier shows a less clear pattern in aspect sensitivity; however, some differences are notable. Cast shadowing appears to be most significant for lower and mid-elevations on north- and south-valley aspects (Fig. 5). Additionally, the upper elevations for the south direction show a large increase in cast shadows. Shaded relief also shows a larger mean change in upper elevations for north and south directions. One interesting point when comparing both glaciers side-by-side is that the impact of shaded relief only surpasses cast shadowing in the north-facing direction. Additionally, in these scenarios, cast shadows and combined shading are largest when the glaciers are in a south-valley orientation.

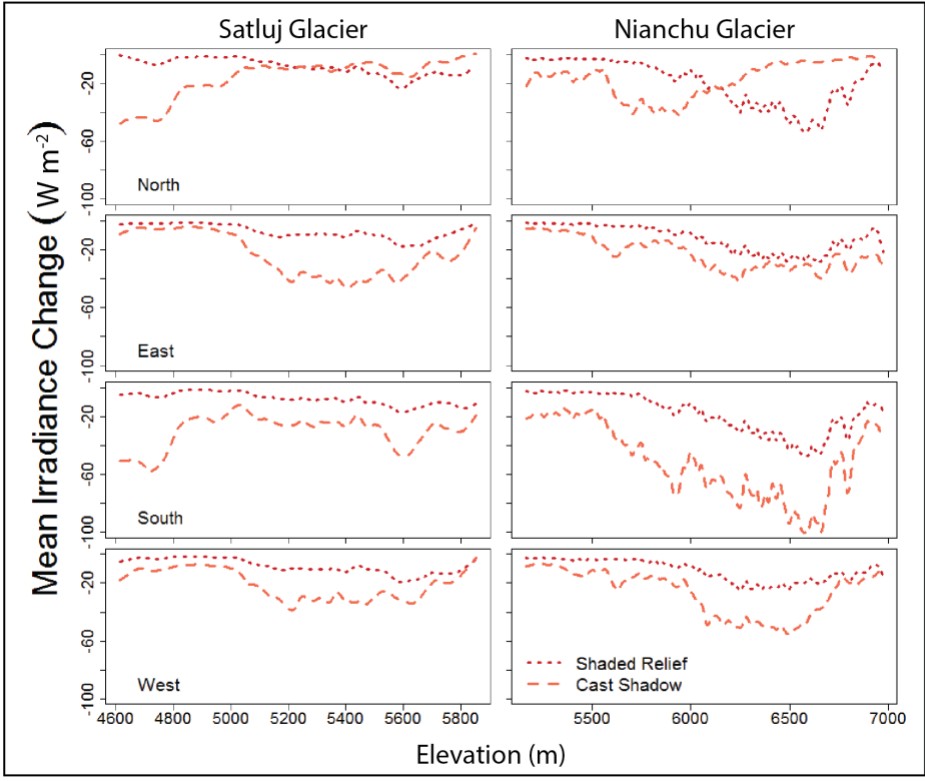

**Figure 5. Idealized scenarios for the Satluj and Nianchu Glaciers. Mean change in irradiance throughout the summer melt season due to shaded relief and cast shadows. Glaciers have been rotated in the four main cardinal directions to observe how shading changes with differing valley aspect. The difference between shaded relief and cast shadows is largest at lower and mid elevations for the north and south valley direction. Also note that shaded relief is only greater than cast shadowing in the upper elevations of the north-facing direction for both glaciers.**





In general, we see that topographic shading is more significant on glaciers with a north- or south-valley aspect, particularly in the lower elevations of valley glaciers. This is of particular interest for regions like the Himalayas where valley glaciers are dominantly north- or south-facing. Although we continue to see variability in shading specific to each glacier, this general result can be a useful proxy to determine how impactful shading might be on a given glacier.

When moving these same glaciers across varying degrees of latitude we see a relative decrease in mean irradiance with increasing degree of latitude for both the Satluj and Nianchu Glaciers (Fig. 6). The trend of relative change in mean irradiance for each glacier closely follows a simple mathematical trend of one minus the tangent of the change in degree latitude (β). Cast shadows become larger with increasing zenith angles, and the zenith angle will increase proportional to latitude. While this simple mathematical trend captures the over-arching changes in irradiance at differing latitudes, it over-predicts for the Satluj

and under-predicts for the Nianchu. This may be due to the fact that lower elevations on the Satluj are surrounded by steeper topography.

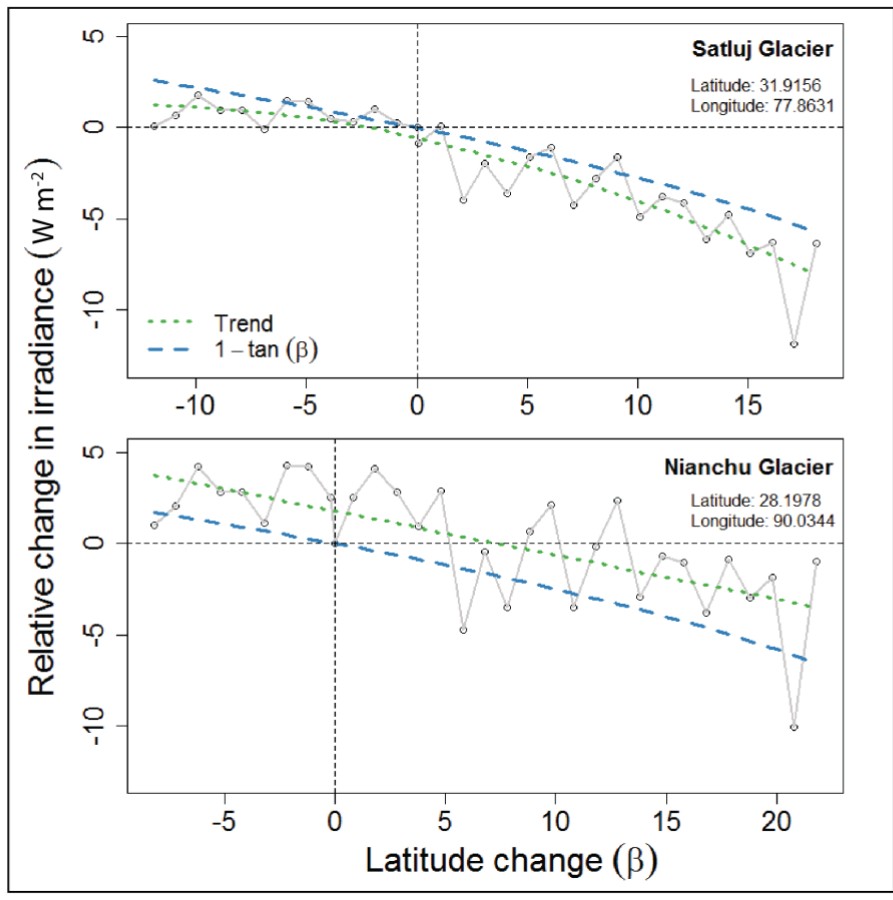

**Figure 6. Idealized scenarios for the Satluj and Nianchu Glaciers. Relative change in mean irradiance due to topographic shading as glacier is moved up and down in latitude., both spanning latitude from 20-50⁰. Mean values are connected (grey) to show**

**stochasticity. The overall trend (green dotted) shows that the effects of shading increase with increasing latitude (β). The relative change mirrors the function 1-tan(β) (blue dashed) which underestimates the relative change for the Satluj, and overestimates for the Nianchu Glacier.**



## 3.3 Regional analysis

Outside of latitude and aspect, the impact of topographic shading is determined by combined characteristics of the immediate surrounding topography. As such, unique basin morphology gives rise to significant variability in the impacts of shading on direct solar radiation across glaciated regions. We apply the topographic models to the full Satluj basin including the

5   surrounding glacial valleys, seen in Fig. 7. This expanded view offers a variety of valley aspects and different morphologies, and a means to explore the spatial variability in the shading components. A hillshade is used to visually enhance topography.

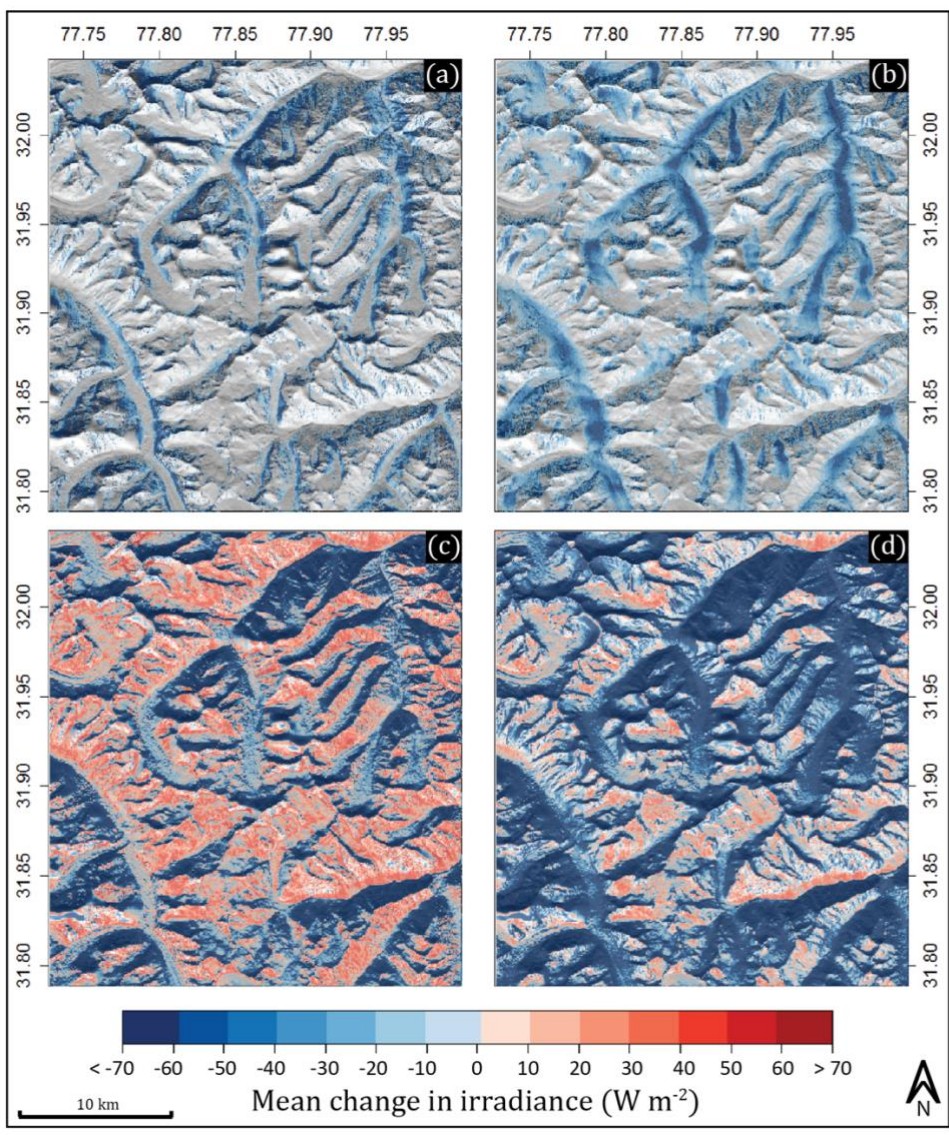

**Figure 7. Expanded view of the Satluj Basin including surrounding glacial valleys of various orientation and aspect. Mean change in irradiance due to shaded relief (a), cast shadows (b), slope and aspect (c), and combined topographic parameters (d) are compared.**

10   **Cast shadows show to be most influential in north- and south-facing valleys and can even offset the increase in irradiance due to slope and aspect for some south-facing valleys. Changes less than ±10 W m$^{-2}$ are not shown.**





Shaded relief (Fig. 7a) appears to be significant only on extremely steep terrain that is not south facing. Although this may
be a significant topographic parameter in the upper cirques of some glaciers, the impact is minimal in glacier valleys, and does
not play a large role in the ablation zones where differences in solar irradiance can impact the mass balance. Cast shadows
(Fig. 7b), on-the-other-hand, are most pronounced in low-elevation north- and south-facing valleys, as predicted in our aspect

sensitivity tests. Interestingly, although the mean change in irradiance increases due to slope and aspect for some south-facing
valleys (Fig. 7c), this effect is offset, and in some cases overwhelmed, by cast shadows (Fig. 7d). By calculating the influence
of these topographic parameters on a larger scale, we are able to confirm the results from our aspect sensitivity analysis in Fig.
5. Spatial patterns throughout the basin show that cast shadows significantly impact mean daily irradiance along deep north-
and south-valley aspects.

We also apply these models to four glaciated regions spanning large portions of the greater Himalaya (Fig. 2), to further test
our idealized scenarios and assess the degree of spatial variability in shading on valley glaciers. Again, we see that North/South-
facing glaciers are more impacted by cast shadowing than East/West-facing glaciers (Fig. 8). Additionally, we see that the

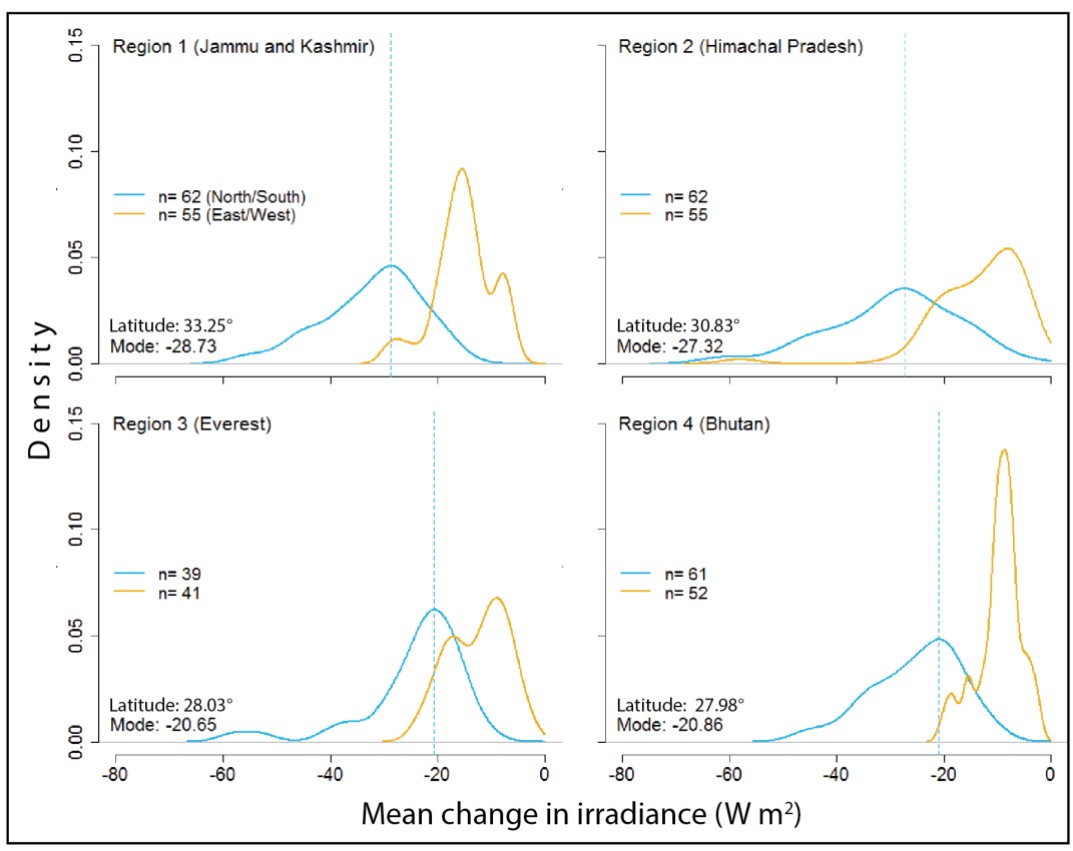

**Figure 8. Regional analysis spanning glaciated basins throughout the greater Himalaya of HMA. A mean value for change in**
**irradiance due to cast shadowing is calculated for the ablation zone of each glacier within each region. We see that the distribution**
**for North/South glaciers are more impacted by cast shadowing than East/West glaciers. Only values below mean glacier elevation**
**were used to calculate the mean change for each glacier. Additionally, we see that the peak value for mean change in irradiance is**
**greater in regions of higher latitudes (Regions 1 and 2).**





peak values in our North/South distributions (mode) of solar irradiance changes show a greater change in regions of higher latitude and relief (Regions 1 and 2), fluctuating from a mean glacial change of -28.7 Wm$^{-2}$ in the highest latitude to -20.7 Wm$^{-2}$ in the lowest. Although we see significant variability between regions, there is generally less spread for East/West- than North/South-facing glaciers. Our regional model validates the findings in our idealized scenarios, confirming that the impact

of topographic shading is more prominent on north- and south-facing valleys with a general increase in the mean change in irradiance between regions located at higher latitudes.

## 4   Limitations and future work

The results of our topographic shading model highlight the important role of topography on the direct clear sky solar radiation. However, because our results rely heavily on simulating topography from digital elevation models, we are limited by the

accuracy and resolution of these DEMs. Although this doesn't diminish our results, we recognize the value in assessing the variability and uncertainty associated with using differing DEM products and higher resolution. Additionally, our results motivate further investigation into the relationship between topography and additional components of both global radiation and energy balance. Finally, the results here are potentially relevant for quantifying the sensitivity of glacier mass balance to climate change.

### 4.1   DEM accuracy and resolution

ASTER GDEM, used in this study, builds surface elevation using orthorectification of two ASTER images, producing a product near 30 m resolution. Although ASTER provides higher detail and generally good elevation accuracy, orthorectification tends to fail in completely snow-covered regions as orthoimages become difficult to align. These issues, along with the resolution, introduce some inaccuracy as flat surface features become exaggerated and ridgelines are smoothed,

which can artificially increase or lower the effects of shading from topographic features (Arnold et al., 2006).

Hopkins (2010) found a linear increase in glacier melt as DEM resolution decreases from 1 to 1000 m. This is due to decreased textural relief as DEM resolution is reduced. In certain scenarios, Hopkins found that total melt increased with DEM resolution by 4%; however, this value only includes shaded-relief and not cast shadowing. In order to illustrate the potential impact of DEM resolution on topographic shading, we calculate a mean glacial change in irradiance as we systematically

decrease the ASTER GDEM resolution. Figure 9 shows a significant decrease in accurately calculating topographic shading as resolution declines for the Satluj and Nianchu glaciers. The initial decline in shading accuracy is abrupt and continues to decrease for both glaciers until the estimated value is nearly 50 % lower than the initial. Topographic shading, particularly from cast shadowing, relies on the ability to simulate a change in energy based on detailed features of the surrounding topography. As detail degrades, so does the true effect of topographic shading. Higher resolution and higher accuracy DEMs,

therefore, will significantly improve the accuracy of the shading models. This begs the question as to whether the impact of modeled shading would continue to increase at the same rate when using DEM resolutions higher than 30 meters.



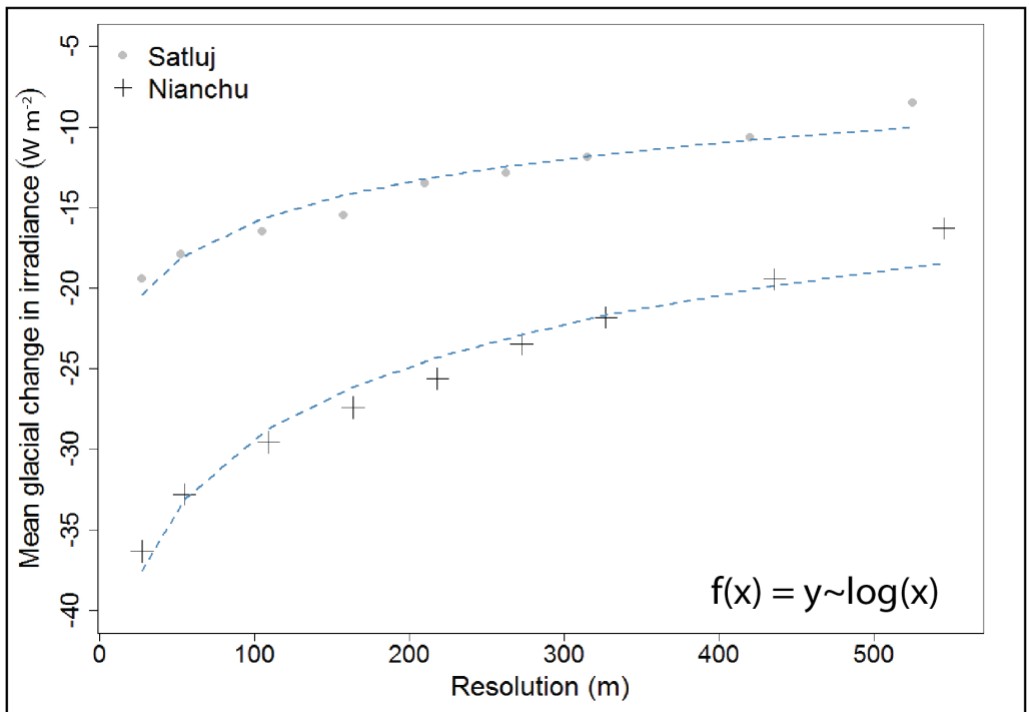

**Figure 9. Showing the effect of decreasing resolution on the change in irradiance due to topographic shading for the Satluj and Nianchu Glaciers. Mean change in irradiance appears to decrease with the natural log of resolution. This is due to smoothing topography.**

## 4.2 Diffuse and terrain-reflected radiation

We focus specifically on direct solar radiation in this study, which is the major component of the global radiation. However, other global radiation terms are also impacted by topography as well. While surrounding topography decreases the amount of direct clear-sky irradiance received at the surface, diffuse sky radiation ($D_s$) and reflected radiation from surrounding terrain ($D_t$), can also alter the net global radiation. For example, Arnold et al. (2006) calculates the total diffuse radiation as:

$$Q_{dif} = F_s D_s + D_t \tag{7}$$

where reflected radiation from surrounding topography (Dt) is calculated as:

$$D_t = \alpha_t (1 - F_s) Q_{global} \tag{8}$$

where $\alpha_t$ is the mean albedo of the surrounding terrain, $F_s$ is the sky-view factor, and $Q_{global}$ is measured global radiation. From Eq. (7) and (8), it is apparent that diffuse sky and terrain-reflected radiation have an inverse relationship with the sky-view factor, which is directly related to the height of the surrounding topography. As such, we would expect the effects of these two components of global radiation to somewhat offset one another in the presence of topography. In Eq. (2), daily mean



direct solar radiation decreases in the presence of higher topography; diffuse sky will also decrease as the view factor decreases, and terrain-reflected will increase the radiation received at the surface.

Depending on the orientation of topography, local meteorological conditions, and the mean albedo of the surrounding terrain, terrain-reflected radiation could significantly offset the effects of topographic shading on global radiation received on

a glacier surface. The relationship between direct, diffuse sky, and terrain-reflected radiation should be further investigated, especially for valley glaciers in steep terrain.

### 4.3    Glacier response to climate change

Most glaciers, under current trends of increasing global temperature, are expected to thin and retreat in response. As they do so, they will thin and retreat into varying amounts of topographic shading. For example, the Satluj Glacier will likely thin and

retreat up-valley. However, this response may be dampened somewhat by the decrease in incoming solar radiation as a portion of the ablation zone resides in the area of maximum shading (Fig. 3). Ultimately, once the glacier tongue has receded beyond the most-shaded region in Fig. 3, the ablation zone will receive a higher amount of direct solar radiation and the mass loss will increase. The link between climate, glacier dynamics, and shading is also relevant for the Nianchu Glacier. As the ablation zone moves up valley into regions of increased shading, the surface will be less affected by direct solar radiation throughout

the day, reducing the melt rate and dampening the response to changes in climate. These scenarios suggest that shading may not only be a useful tool for improving our understanding of the current mass balance of valley glaciers throughout HMA, but could likely improve our understanding of the magnitude of glacier response to climatic change under future climate scenarios. Similarly, the variability in shading on a glacier surface over time is likely to play a role in explaining glacier sensitivity to historical climate changes. Future work should focus on quantifying the role of topographic shading in glacier response to

climate changes.

### 5    Summary and conclusions

Topographic shading is comprised of two components: shaded relief, and cast shadows. Shaded relief is due to slope and aspect and occurs when a surface is blocked from the sun's rays due to its own relief. Cast shadows occur when solar rays project shadows from one topographic feature onto another, which occurs commonly in valleys surrounded by steep valley walls.

We create a topographic solar radiation model in order to quantify the effects of topographic shading on valley glaciers with a variety of aspects and latitudes, and within a range of terrain settings. We find that potential direct clear-sky solar radiation on a glacier surface can be significantly affected by shading in regions of steep topography and high relief such as is common in HMA. Overall, there is an increase in shaded relief with increasing elevation, as slope angles generally become steeper. In contrast, cast shadowing does not show any clear trend with elevation; rather, it appears to be controlled by the distance and

direction of the immediate surrounding topography, as well as size of nearby terrain. This makes cast shadowing extremely variable along and between glacier valleys. Importantly, we see that cast shadowing can account for a significant decrease in





irradiance in the glacier ablation zone. Indeed, cast shadowing is typically the dominant mechanism contributing to total shading in all but the steepest of slope angles, as such, it should be incorporated in energy balance models.

We find that glaciers with north- and south-valley aspects are generally more influenced by cast shadowing. Additionally, we see a general increase in shading with increasing latitude. This suggests that parameters such as latitude, aspect, slope and
others may be useful in predicting the overall effect of shading for a particular glacier. This could be useful in estimating the impact of shading on a regional scale in order to incorporate the full impact of topography in large-scale models. Furthermore, DEM spatial resolution noticeably alters the modeled accuracy of topographic shading on a glacier's surface.

We show that topographic shading results in a significant mean change in irradiance, particularly in the ablation zone of north- and south-facing valley glaciers. This implies that topographic shading has the potential to greatly impact the surface
energy balance of Himalayan glaciers during the summer melt season, and could also be useful in understanding different glacier mass balance estimates throughout HMA.

*Acknowledgements*.   We acknowledge the support for this research through funding awarded to Summer Burton Rupper (NSF EAR 1600587 and NASA 15-HMA15-0030). We also acknowledge the valuable constructive feedback from Drs. Rick Forster
and Simon Brewer, as well as Eric Johnson, Jewell Lund and others from the University of Utah.

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
