# Peer review of "Impacts of topographic shading on direct solar radiation for valley glaciers in complex topography"

_The Cryosphere, 2018_

## Referee Comment (RC1) · G. Evatt (Referee) · 1 Jun 2018

This paper presents a reminder to people conducting glacier energy balance models, about the impact topographic shading (from both the surrounding terrains and self-shadowing) has upon SW energy receipts. By using a computer program coupled with elevation data, they calculate the degree to which topographic shading effects two distinct glaciers. The paper is generally well written and presents an interesting flag of how important topographic shading is. My main concern is that this point is reasonably well known, and even if it were not, the methodology presented here does not present an easily replicable rectification. And as the authors say at the end of the manuscript,

future work should consider terrain-emitted/reflected energy; in my view this paper should also have considered this, for the decrease in SW energy from shading is partly offset by the increase in topographic reflectance/LW energy emittance, and thus the degree to which shading effects the surface energy balance is not established.

Main points:

Novelty: Whilst true that people have overlooked the precise role topographic information has on SW fluxes, most of the work I am familiar with uses field data for calibrating SW fluxes. In so doing, this already has the topographic information built in. This is less the case if one is purely using a computational approach with no in-situ field data. But if one is doing that, then one certainly also needs to worry the size of offset from SW reflection from terrain and the associated LW fluxes, which this paper does not.

Replication/utility: Equations 3-6 need much more explanation and certainly a diagram, showing where all the angles/fluxes are acting. In terms of replication, it would have be very useful to see how these quantities (or the fluxes they they then produce) compare when one using mean topographic information. Mean topographic information is much easier to estimate/use than using exact elevation data. A comparison between the exact and mean fluxes would then be very useful: if close, then that gives people an easy fix; if quite different, then it shows a fine grained approach, as advocated by this paper, is required.

Minor points:

-P3 l 10. Here, and elsewhere in the paper, the use of 'slope and aspect' is a little confusing. As the authors say, cast shadows contain both slope and aspect already, so what exactly does the different 'slope and aspect' refer to? A diagram would be nice, for these distinctions need to be effortlessly clear.

Eqn 4, missing a dt?

p9, l5, change 'won't' to 'will not'.

---

## Referee Comment (RC2) · Anonymous Referee #2 · 15 Jun 2018

I agree with the review comments that have already been made. This isn't a new insight but it is helpful to be reminded of its potential significance. The modelling section needs to be clarified - equations (3) - (6) are not easy to understand as they are, and would certainly benefit from diagrams to show how things are defined. (And as a minor comment, the lower limits for the integrals in these expressions have to be zero, not 1, surely?) The use of the secant approximation for the air mass (eq. 1) is discussed, and the fact that it will introduce bias into the results is correctly identified, but I was left wondering why the authors didn't simply fix the problem with a better approximation. As the previous reviewer, I feel that the lack of explicit consideration of diffuse radiation is also a significant omission.

[Figure]

The discussion of the impact of DEM resolution in section 4.1 is certainly relevant, but needs to acknowledge the fact that, while the sampling interval of GDEM is 30 m, its actual spatial resolution is coarser than that. There is relevant literature on this, e.g. Hengl & Reuter 2011 in general, Rees 2012 for polar environments, no doubt others, showing that the actual spatial resolution is more like 100 m so that some at least of the trend noted in fig 9 can be attributed to the characteristics of the DEM not the terrain.

---

## Author Comment (AC1) · 31 Aug 2018

**Author's response to reviewer comments: Impacts of topographic shading on direct solar radiation for valley glaciers in complex topography**

First, we would like to thank The Cryosphere for their unique open review process, and in particular, our editor for extending the deadlines for this paper. We sincerely appreciate the comments and suggestions received from Geoffrey Evatt, and Anonymous Referee #2. We address each of the comments below, and believe the paper is significantly improved because of their feedback.

Reviewer Comments: **Bold**
Author's Response: AR: Regular text
Added Text: Blue

**Response to Referee #1, Geoffrey Evatt**

Overall concerns:

1. **The paper is generally well written and presents an interesting flag of how important topographic shading is. My main concern is that this point is reasonably well known, and even if it were not, the methodology presented here does not present an easily replicable rectification.**

AR: Although we agree that the importance of topographic shading is generally well known, it is often excluded or oversimplified in many glaciology studies. Additionally, the magnitude of impact topographic shading can have on the direct solar radiation and overall energy balance of a glacier has only narrowly been discussed in detail, and has not been addressed in the uniquely complex topographic region of the Himalayas. This paper aims to clarify any misunderstandings regarding modeling methods for shading, show the magnitude of importance shading can have, and generate ideas into further improving the accuracy of surface energy balance models in complex terrain.

In order to improve the replication of this approach, we clarify the model and equations in the text and create a new diagram (Figure 3). See following comments.

2. **The authors say at the end of the manuscript, future work should consider terrain-emitted/reflected energy; in my view this paper should also have considered this, for the decrease in SW energy from shading is partly offset by the increase in topographic reflectance/LW energy emittance, and thus the degree to which shading effects the surface energy balance is not established.**

AR: In section 4.2 we explain the importance of including these energy fluxes in future work, as noted by the reviewer. However, we disagree that these additional components are required to address the impacts of topographic shading. The goal of this paper is to explore the magnitude

to which direct shortwave radiation may be altered if topography is not incorporated correctly, as has historically been the case for many glaciological studies. Direct shortwave radiation is by far the largest energy flux during the summer months, and is directly affected by shading from surrounding terrain. While it is important to include other energy fluxes for a complete surface energy balance, it is beyond the scope of this paper and would not improve our understanding of the role of topographic shading on direct solar radiation. As such, we feel that this paper includes valuable information and discussion regarding the impact of topography as it relates to direct solar radiation.

Main points:

1. **Novelty: Whilst true that people have overlooked the precise role topographic information has on SW fluxes, most of the work I am familiar with uses field data for calibrating SW fluxes. In so doing, this already has the topographic information built in. This is less the case if one is purely using a computational approach with no in-situ field data. But if one is doing that, then one certainly also needs to worry the size of offset from SW reflection from terrain and the associated LW fluxes, which this paper does not.**

AR: Calibrating SW fluxes with field data is generally preferred over a solely model based approach. Unfortunately, there is little existing data in remote glacierized regions of the world, particularly in the Himalaya. Due to this fact, the community must rely heavily on elevation data obtained via satellite and modeled climate data in order to estimate the glacier surface energy balance. Additionally, even with adequate field measurements, the spatial coverage is often insufficient to properly determine the fully distributed impact of topographic shading over the entire coverage of a glacier. The topographic information is included at the location on a glacier in which measurements were taken, however, the impact of topography is extremely spatially variable. Figures 3b and 4b show the spatial heterogeneity of cast shadowing across the surface of two different glaciers in the Himalaya, and highlight this point. In addition to improving modeled SW fluxes, incorporating this shading algorithm will indeed improve the accuracy of field calibration data across the surface of a glacier. Models are also requisite for projecting glacier changes under future climate scenarios, as the role of shading shifts over time as the glacier thins and retreats (or thickens and advances). Thus, while field data are invaluable, models are also necessary for many glaciological research questions.

We add discussion in the introduction to the paper at P3:L11 addressing this directly, as this may be a question other readers have as well.
P3:L11– "Improving glacier models is essential in order to predict melt, particularly in remote regions where little data exists. Additionally, point field measurements generally lack the spatial heterogeneity of energy flux values across the surface of a glacier, and consequently require additional topographic information and integrative modelling."

2. **Replication/utility: Equations 3-6 need much more explanation and certainly a diagram, showing where all the angles/fluxes are acting. In terms of replication, it**

**would have be very useful to see how these quantities (or the fluxes they then produce) compare when one using mean topographic information. Mean topographic information is much easier to estimate/use than using exact elevation data. A comparison between the exact and mean fluxes would then be very useful: if close, then that gives people an easy fix; if quite different, then it shows a fine grained approach, as advocated by this paper, is required.**

AR - equations: P5:L15-16 mentions that for Equations 3-6 "We use variations of Eq. (2) in order to determine the daily mean change in solar radiation from specific topographic parameters shown in Fig. 1." The flowchart in the bottom right corner of Figure 1 explains how each term in Equation 2 is associated with each topographic parameters. This equation and terms are well known among individuals interested in energy balance within the glaciology community. Essentially equations 3-6 calculate the difference between Equation 2 and a model that excludes the topographic parameter of interest, which is then integrated throughout the day.

We have added an additional diagram (Figure 3) to illustrate the derivation of Equations 3-6 and define the topographic parameters of interest. Text is also added at P5:L27.
P5:L27 - "Figure 3 shows the derivation of these equations with respect to Equation 2 (Model 1), and illustrates the topographic parameters of interest. In order to parse out the influence of each parameter, a second model is created excluding the parameter of interest. Equations 3-6 calculate the difference between Equation 2 (Model 1) and a model that excludes the topographic parameter of interest, which is then integrated over the course of a day. The result of these equations is a change in irradiance due to a specific topographic parameter."

[Figure]

**Figure 1. Diagram illustrating the derivation of Equations 3-6. Model 1 (same as Equation 2) is the base model and incorporates both methods of topographic shading (S) and the effect of slope and aspect ($\theta_i$). Additional models are also created, each excluding some component of topography. The difference between Model 1 and Model 1a shows the change in irradiance due to slope and aspect on the surface of the glacier. Model 1b calculates the sum of irradiance that would arrive on a flat surface at locations on the glacier where the incidence angle ($\theta_i$) is greater than 90°. Model 1b is the only scenario not dependent on Model 1. The difference between Model 1 and Model 1c shows the change in irradiance due only to cast shadows. Model 1d removes surrounding terrain and assumes the glacier surface is a flat plane, the difference between this and the original model shows the combined effect of removing all topographic information from the DEM.**

AR - replication: Rather than compare our fluxes against the result of a mean topographic proxy, we opted to focus on the impact of DEM resolution in section 4.1 and Figure 9. These results show that in regions of complex topography, topographic effects can drastically alter energy fluxes. As such, some amount of distributed elevation data is much more useful than mean topographic information. Due to this fact, we considered the question of DEM resolution to be a more pertinent comparison than mean topographic fluxes.

Minor Points

1. **P3 L10. Here, and elsewhere in the paper, the use of 'slope and aspect' is a little confusing. As the authors say, cast shadows contain both slope and aspect already, so what exactly does the different 'slope and aspect' refer to? A diagram would be nice, for these distinctions need to be effortlessly clear.**

AR: The vocabulary used to describe the topographic parameters in this paper is very literal of the physical processes altering the energy arriving at the surface. As such, the term 'slope and aspect' refers to the slope and aspect of the surface, which we include as one of our topographic parameters altering the amount of irradiance on a glacier. P3:L12-14 states that we include this parameter as a comparison against the impact of our two shading methods. However, previously on P2:L2-4, we note that shaded relief also relies on information about the slope and aspect of the surface in order to determine whether a given cell is in shade. Figure 1 illustrates the two main types of shading, shading based on the relief and positioning of the surface topography (shaded relief), and shadows cast from the surrounding terrain (cast shadows). Although shaded relief relies on obtaining information about the slope and aspect, it is still a component of topographic shading. The reviewer claims that the paper states, "cast shadows contain both slope and aspect already," We are not positive where this reference comes from and do not believe such a statement exists, as it would be an incorrect statement. However, we believe that the confusion is between 'slope and aspect' as a separate parameter, and the fact that information about slope and aspect is used in determining areas of shaded relief.

We added text at the end of the paragraph at P3:L14 defining the term 'slope and aspect' and edited the caption on Figure 1 to include information about this term and the other topographic parameters of interest. Additionally, we feel that Figure 3 in conjunction with Figure 1 now clarifies any other confusing terminology throughout the paper. We thank Referee 1 for helping us see that this needed further clarification.
P3:L14 - "Hereafter, 'slope and aspect' will refer to a topographic parameter responsible for changing irradiance values based on the orientation and slope of the surface in unshaded areas of the glacier."
Figure 1 caption addition - "Red coloring indicates areas 'in sun' that are impacted by a change in irradiance due to the slope and aspect of the surface. Cast shadow, shaded relief, and 'slope and aspect' are the three topographic parameters compared in this paper."

2. **Equation 4, missing a dt?**

AR: Yes, thank you for catching that.

3. **P9:L5, change 'won't' to 'will not'.**

AR: Changed, much appreciated.

**Response to Anonymous Referee #2**

Overall concerns

1. **I agree with the review comments that have already been made. This isn't a new insight but it is helpful to be reminded of its potential significance. The modelling section needs to be clarified - equations (3) - (6) are not easy to understand as they are, and would certainly benefit from diagrams to show how things are defined.**

AR: This was addressed in the previous review response. A diagram (Figure 3) has now been included to aid in explaining the equations.

Major points

1. **The use of the secant approximation for the air mass (eq. 1) is discussed, and the fact that it will introduce bias into the results is correctly identified, but I was left wondering why the authors didn't simply fix the problem with a better approximation.**

AR: We have removed this sentence from the paper due to the fact that this bias is considered to be insignificant to our results and detracts from the main goals of this paper.

2. **As the previous reviewer, I feel that the lack of explicit consideration of diffuse radiation is also a significant omission.**

AR: This was addressed in the previous response under section 2 of the Overall Concerns.

3. **The discussion of the impact of DEM resolution in section 4.1 is certainly relevant, but needs to acknowledge the fact that, while the sampling interval of GDEM is 30 m, its actual spatial resolution is coarser than that. There is relevant literature on this, e.g. Hengl & Reuter 2011 in general, Rees 2012 for polar environments, no doubt others, showing that the actual spatial resolution is more like 100 m so that some at least of**

**the trend noted in fig 9 can be attributed to the characteristics of the DEM not the terrain.**

AR: Reviewer 2 points out the issues in spatial resolution associated with the ASTER GDEM. Section 4.1 advocates the need for higher spatial resolution when determining the impact of topographic shading. Figure 10 shows that the influence of shading diminishes as resolution decreases, assuming the provided resolution of the GDEM. Although the actual spatial resolution might be less than the sampling resolution of the terrain, the coarsening of the DEM in Figure 10 illustrates that spatial resolution (actual or sampled) affects the results. As this is largely a theoretical study, we feel that this issue is less of a concern, however, we agree that this information is useful to other readers.

We added discussion in section 4.1 at P15:L20 addressing possible issues with the GDEM, as this could be useful information for those interested in incorporating this research into their models.
P15:L20- "It should also be noted that although the sampling interval of the GDEM is 30 m resolution, the spatial resolution may be up to 3 times coarser in some regions (Hengl and Reuter, 2011). Despite the shortcomings of this DEM product, the findings in this study still demonstrate clearly the impacts of DEM resolution on incoming shortwave radiation."

Minor Points

1. **And as a minor comment, the lower limits for the integrals in these expressions have to be zero, not 1, surely?**

AR: Thank you for catching that. We have corrected the limits for the integrals on equations 3-6.

---

## Author Response (AR2)

**Author's response to editor and reviewers: Impacts of topographic shading on direct solar radiation for valley glaciers in complex topography**

We appreciate the additional feedback from both reviewers and suggestions from the editor. Once again, we believe that the paper has benefitted because of this feedback. For each comment we briefly provide a response (AR) which is often followed by explicit changes made in the paper (added/edited text appears blue). Furthermore, based on this feedback we have made some additional small changes to the paper in order to clarify confusing sections and improve flow. These changes mostly relate to restructuring parts of the key motivating paragraphs in the introduction, and include: moving general statements in the motivation to earlier paragraphs (P2:L16-17), slight modifications to the references listed on P2:L19-21 and P3:L9-11, and moving the technical description of slope and aspect from the motivation to the data and methods section (P4:L8-12). We also changed figures 1 and 3 to include the word component instead of parameter.
The marked-up manuscript can be found after the response to review #2.
(Page and line numbers reference the edited markup document)

Reviewer Comments: **Bold**
Author's Response: AR: Regular text
Added/Edited Text: Blue

**Editor Suggestions**

1. **I suggest that you clarify some of the technical issues pointed out by reviewer #2…**

AR: Technical issues have been clarified (see section for referee #2).

2. **…and that you also add a few sentences about the significance of the work. I think it is acceptable if the significance is incremental at best, as suggested by one of the reviewers, if this is acknowledged in the text. This can be supported by the requested list of other studies with similar focus. On the other hand, if you explained more explicitly that this is a case study illustrating again the importance of shadowing that would be acceptable as well.**

AR: We have done a major rewrite of the key motivation paragraphs in the introduction (P2:L13–P3:L29) that discuss previous studies related to shading, and places this study into better context of the unique contributions being made. This is discussed in detail in comments 1 and 2 for referee #1.

**Referee #1**

1. **The authors do not act upon the suggested mains edits, and instead repeats the thrust of the paper. (I fail to see how the Himalaya is unique in regards their work. Unless they mean their approach has already been used elsewhere, but not yet in the Himalaya -in which case their study seems quite incremental).**

AR: We have rewritten the end of the introduction (P2:L13–P3:L29), which provides motivation for this paper, and describes previous studies and relevant background information regarding topographic shading, Additional detail describing this section and the significance of this paper can be found in the next comment.

In order to clarify the underlying motivation and significance of this paper we have done a major revision of the final paragraphs in the introduction on P2:L19–P3:L29 in order to illustrate these main 6 points:

1. Several studies explicitly incorporate shading correctly, however, only a handful of known studies observe the direct impact of shading for a single glacier.
2. Varied results from these studies have created inconsistent interpretations of the importance of shading in energy balance.
3. Many other studies claim to incorporate topographic shading, but do so incorrectly, leading to erroneous results.
4. No studies exist that explore how topographic shading changes over variable complex terrain, against glacier aspect, and across differing latitude.
5. To date, no systematic studies have been performed that quantify the magnitude and variability of errors due to incorrectly incorporating topographic shading.
6. We specifically choose HMA because of the varied complex topography, little to no previous investigation of topographic impacts throughout the region, and the fact that modeling is crucial in HMA due to the remote nature of these glaciers.

See comment 2 for revision text.

**2. Within the text, the authors should give a clear and exhaustive list of studies where shadowing has been overlooked to a detrimental degree. At present their claim in regards the oversight is not given any supporting evidence.**

AR: P3:L9-11 lists seven prominent studies (ranging 1997-2016) which claim to incorporate topographic shading, but exclude the impact of cast shadows. This list is not comprehensive as many studies do not explicitly state the method used for incorporating shading. We have listed only studies that clearly model this component incorrectly in order to illustrate that there is still confusion within the community. Also, these incorrect approaches are still being both cited and used in new studies, propagating the problem.

We also list eight highly cited studies (P2:L19-21) which explicitly incorporate topographic shading correctly and allude to the potential importance. However, only a few of these studies focus some attention directly on specifying the impact of shading for a single glaciers. Unfortunately, these results have some inconsistencies between each other. The impacts of shading on a single glacier also limit the transferability of results to other regions. To date, no systematic studies have explored shading over varying regions of complex terrain, variable aspect, and differing latitude. We therefore consider the relevance of this paper to be useful in adding to the current understanding of topographic shading, and relevant in terms of location, as few studies have directly focused on this component in HMA

We have rewritten the key motivating paragraphs at the end of the introduction (P2:L19–P3:L29) in order to more clearly state the motivation for this paper.

P2:L15–P3:L29 – Major revision (This section is large so only the first and last lines are included here) "Several glacier-related studies explicitly model topographic shading and allude to the potential importance…This study focuses on improving the current understanding regarding the impact of both shaded relief and cast shadowing on glaciers in the complex and varied topography of HMA."

3. **They may also wish to clarify some terminology for the reader, as 'self-shadowing' already seems to be commonly used.**

AR: Both "shaded relief" and "self-shadowing" have been used in the literature. Our reason for choosing the former is due to an apparent larger occurrence, although, as pointed out by referee #1, "self-shadowing" is also common.

P2:L2-3 – We modified the introduction of shaded relief and included other common terminology: "Shaded relief (also referred to as self-shadowing) occurs when a given location is obscured by the sun solely due to the slope and aspect of the terrain."

**Referee #2**

1. **In fig 7, and the text describing it, I don't understand why a dependence on 1-tan(beta) would be expected, or how the dashed lines can represent such a variation if they both pass through zero when beta=1 (which they appear to do).**

AR: We thank referee #2 for pointing out the error in the equation for figure 7. We have corrected the '1-tan(β)' relationship to '-tan(β) dβ.' P12:L12-16 explains the general observed relationship between irradiance and latitude, however we agree that this section needs additional explanation for the mathematical relationship. We have rewritten part of this section to include additional text that more directly explains our findings in figure 7:

The equation in figure 7 has been corrected from '1-tan(β)' to '-tan(β) dβ'

P13:L13-19 – Added text explaining the simple mathematical relationship: "This mathematical trend *-tan(β) dβ* arises from a simple trigonometric relationship between the zenith angle (Z) and the amount of cast shadows (S) for topography of a given height (H). As the zenith angle increases, the length of shadows cast also increases *tan(Z) = S/H*. Apart from variation over the course of a day, zenith angles increase proportional to an increase in latitude (Iqbal, 2012). As such, we assume that the change in irradiance from cast shadows across changing latitude, for a glacier with fixed topography, is largely a function of  *-tan(β) dβ*, with variation based on changes in the intersection of azimuth angles and surrounding topography."

2. **Fig 10 presents the effect of DEM resolution on topographic shading, and suggests a logarithmic relationship. I don't really follow what is being proposed here. The**

equation 'f(x)=y~log(x)' is unclear. I guess it means either that $\Delta I = -a \log(r)$ or $-a \log(r) + b$ (I being irradiance and r resolution), though whether a or b should be the same for the two glaciers of not is not discussed. The dotted lines in the graph are not explained, but perhaps least-squares fits to the model (although they appear to consist of straight-line segments rather than smooth logarithmic functions). But most puzzling, I don't know how to interpret the fact that these models will not work as the DEM resolution tends to zero.

AR: Figure 10 shows that increased resolution reduces the change in irradiance due to topographic shading over a given glacier. Although some segments appear almost linear, the largest and most relevant changes appear between ~30–100 meter resolution, suggesting that the relationship is not linear. The basic equation 'f(x) = y~log(x)' is not a full accounting of the relationship between resolution and the calculated change in irradiance from topographic shading, however we feel that the blue line (which shows the change in irradiance in regard to the natural log of resolution) is useful to illustrate the initial dramatic loss of information when resolution changes from ~30 to ~100 meters. We believe that figure 10 is still useful in illustrating the potential bearing of DEM resolution on modelled shading. In regards to the model approaching zero, we assume that at some point the mean glacial change in irradiance would level out.

We have removed the equation in figure 10 as it is not a full accounting of the relationship between resolution and topographic shading. However, we leave the trend (blue line) showing the abrupt loss in mean glacial irradiance at finer resolutions due to topographic shading.

We added/edited the figure caption text: "…The mean change in irradiance appears to decrease with the natural log of resolution (blue line). Although this does not fully describe the relationship observed, it aids in highlighting the non-linear change in irradiance with respect to resolution. This is due to smoothing topography as resolution decreases."

P17:L24-27 was also modified: "As resolution coarsens, the modelled mean glacial change in irradiance abruptly decreases (a near 17 % loss for the Nianchu as resolution increases from 30 m to ~100 m) and continues to steadily decrease for both glaciers until the modelled change in irradiance is nearly 50 % below the initial."

**Minor:**

**p1L7 'shading' is not a parameter (though it can be parameterised) Components???**
We have changed 'parameter' to 'component' throughout the paper

**p1L14 'some predictable characteristics' - seems a misleading expression here: surely it is entirely predictable if topography is known?**
Changed to "can be predictable"

**P1L19 'thermal expansion' of oceans, presumably**
Changed to "after thermal expansion of oceans"

**p3L5 'he suggested' -> 'they suggested'**
Changed to "they"

**p4L3 'parse out' does not seem like a correct phrase here. I only know its meaning in the contexts of language and computer programming. Perhaps 'attribute'?**
Changed to "attribute"

**P5L26 'parse out' again**
Changed to "determine"

**p7L25 'according to the best perceived aspect' - I don't know what this means.**
Changed to "based on visual inspection"

**p14L9 delete hyphens from on-the-other-hand**
Deleted

**p16L20 a linear increase in MODELLED glacier melt?**
Added "modelled"

**p18L10 omit hyphen from 'most-shaded'**
Deleted

**omit the dots from equations 3-6; don't indent the text immediately following an equation (unless a new paragraph)**
Deleted; Removed indentation for after equations 1 and 2

[revised manuscript text omitted]